# Self-Reported Parental Interactions through Play with Young Children in Thailand: An Analysis of the 2019 Multiple Indicator Cluster Survey (MICS)

**DOI:** 10.3390/ijerph19063418

**Published:** 2022-03-14

**Authors:** Thitikorn Topothai, Rapeepong Suphanchaimat, Chompoonut Topothai, Viroj Tangcharoensathien, Nisachol Cetthakrikul, Orratai Waleewong

**Affiliations:** 1International Health Policy Program, Ministry of Public Health, Nonthaburi 11000, Thailand; rapeepong@ihpp.thaigov.net (R.S.); chompoonut@ihpp.thaigov.net (C.T.); viroj@ihpp.thaigov.net (V.T.); nisachol@ihpp.thaigov.net (N.C.); orratai@ihpp.thaigov.net (O.W.); 2Division of Physical Activity and Health, Department of Health, Ministry of Public Health, Nonthaburi 11000, Thailand; 3Saw Swee Hock School of Public Health, National University of Singapore, Singapore 117549, Singapore; 4Division of Epidemiology, Department of Disease Control, Ministry of Public Health, Nonthaburi 11000, Thailand; 5Bureau of Health Promotion, Department of Health, Ministry of Public Health, Nonthaburi 11000, Thailand

**Keywords:** child, parents, interaction, play, growth and development, cognitive skills, Thailand

## Abstract

Parental interactions through play contributes significantly to child development of cognitive and executive functioning skills. In Thailand, there is little evidence of factors contributing to parental–child interactions. In response to SDG target 4.2.3 monitoring (the percentage of children under 5 years experiencing positive and stimulating home learning environments), this study aimed to assess the prevalence and profile of parental interactions with their children under the age of five. We analysed data from the 6th Multiple Indicator Cluster Survey (MICS) conducted by the National Statistical Office in 2019. Face-to-face interviews with mothers and/or legal guardians were conducted. A total of 8856 children under the age of five were enrolled in this survey. Most participants, 90.3%, had engaged in at least four out of six activities with their children. Multivariate logistic regression analysis showed that children raised by parents with secondary or post-secondary educations had a significantly greater chance to have parental interactions than children raised by parents who completed primary education (adjusted odds ratio (AOR) = 1.66, and AOR = 2.34 for secondary and post-secondary education). Children who possessed three or more children’s books and had experience of toy play had a significantly higher chance of having parental interactions (AOR = 3.08 for book possessing, and AOR = 1.50 for the experience of toy play). Children who spent 1–3 h daily screen time had a significantly lower chance of having parental interactions than those who spent less than one hour of screen time (AOR = 0.67). In conclusion, with the emerging influence of digital technology, we recommend family and community promote parental interactions through play with young children.

## 1. Introduction

Play is a self-motivated activity that involves active participation and naturally leads to pleasant discovery [1,2]. Play is one of the most important ways in which young children gain essential knowledge and skills; play opportunities and environments that promote play, exploration, and hands-on learning are at the core of effective pre-primary school interventions. Critical skills that children acquire through play in the preschool years form part of the fundamental building blocks of future complex “21st-century skills” [3]. UNICEF describes attributes of childhood play, see Box 1. Through play, children develop a variety of skills. Children can express themselves via play while developing their inventiveness, dexterity, and physical, cognitive, and emotional strength [1,2]. Play is not only about having a good time; it’s also about taking risks, experimenting, and pushing limits [4]. Thus, play is necessary for all children, especially during the digital and technology disruptive era where global citizens require more innovation, more originality, and less conformity [2,4]. 

Box 1Key attributes of childhood play. Source: United Nations Children’s Fund. Learning through play; 2018 [3].Play is meaningful; children play to make sense of the world around them.Play is joyful; often making children smile and laugh.Play is actively engaging where children are deeply involved and combine physical, mental, and verbal engagement.Play is iterative, they practice skills, try out possibilities, revise hypotheses, and discover new challenges, leading to deeper learning.Play is socially interactive which allows children to communicate ideas, to understand others through social interaction, paving the way to build deeper understanding and more powerful relationships.

A systematic review suggests that due to the complexity of dyadic interactions, paternal behaviour can have vastly different associations, both positive and negative outcomes on child development [5]. Stimulating play also enhances the adaptability of a child to a chronic somatic condition such as cystic fibrosis, congenital heart defects, and promotes cognitive, social, emotional, and psychomotor functioning, who otherwise will have a significantly increased risk for physical, social, emotional, and cognitive problems later in life [6]. Though loose parts plays are promising, their impacts on children’s cognitive, social and emotional development need further research [7].

In the UK and elsewhere, modern, urbanization and technologies have significantly reduced child play experiences of free play and unstructured time. Around half of all children worldwide now live in urban settings, and experience very reduced opportunities for outdoor free play in natural environments, due to parental concerns about traffic, ‘stranger danger’, germs and disease [8].

Parents and caregivers play crucial roles in supporting children for the development of their full potential through several activities including appropriate, affective relationships through playing [9,10,11]. Interactions during playtime show youngsters that their parents are paying attention to them, and this helps foster the development of long-term relationships between them [9,10,11]. Parents who get a glimpse into their children learn to communicate with them more successfully [4,12]. Reading, observing, playing with them, and conversing with and listening to them are all examples of favourable ways to respond and interact with children [2,4]. Many inexpensive books, toys (e.g., wooden spoons, blocks, balls, puzzles, crayons, boxes, and basic household items) applied by parents to play with their children encourage children’s creativity [2,13].

Despite multiple benefits of early encounters with quality caregiving through play, some children have been significantly restricted in their amount of play with their parents [4]. Children from low-income households may have little opportunity to play since their parents are too busy due to long hours of work or may have less access to high-quality public places and recreational facilities in their neighbourhoods [10]. Furthermore, as a result of the advent of digital and technology disruption, young children in this generation are exposed to more technology and have access to more electronic devices and may have disproportionately excessive screen exposure [2,14]. 

In Thailand, various research on parental interactions through play have been published. According to a series of reports of the ‘Multiple Indicator Cluster Survey (MICS)’ in 2006, 2012, and 2016 [15,16,17] approximately 79%, 93%, and 93% of children younger than 5 years were engaged in four or more playing activities with caregivers; these were book reading, storytelling, singing, naming or numbering or drawing, outdoor play, and family play. However, there is little evidence of factors associated with parental interactions through play, such as demographic data, play equipment availability, and screen time duration. A better understanding of this relationship will aid policymakers in developing appropriate comprehensive multi-sectoral policies to encourage parental interactions and child development at an early stage. 

The objective of this study was to assess the prevalence and profile of parental interactions with their children and identify household characteristics that influence the parental interactions among Thai children under the age of five.

## 2. Materials and Methods

### 2.1. Study Design, Data Source, and Participants 

A cross-sectional quantitative design was used in this research. The data obtained in this study was one part of the 6th MICS jointly conducted by the National Statistical Office (NSO) and the United Nations Children’s Fund (UNICEF) in 2019 [18].

The Thailand MICS 2019 sample was meant to produce estimates for a significant variety of health indicators related to children and women at the national level, for both urban and rural areas, and five regional domains: Bangkok, central, north, northeast, and south. The major sampling strata were designated as urban and rural areas per province, with the sample being chosen in two stages (enumeration areas [EA] and households). A certain number of EAs were chosen systematically within each stratum, with probability proportional to size. Households (either with or without children under the age of five) were counted after a household list was completed in the designated EAs. In the second stage, a systematic sample of households was picked from each category within the sample EA. At the national level, 1958 sample EAs and 40,660 households were chosen. This study included 8856 children under the age of five (2–4 years) from these households.

### 2.2. Data Collection, Questionnaire Design, and Variable Management

The NSO field crew conducted face-to-face interviews with mothers and/or legal guardians in each household. Each interview lasted an average of 60 min. Data were instantly entered into the mobile tablets by field-trained employees. If the mothers and/or legal guardians in the visited household were not present during the first round of the survey, they were revisited. The MICS questionnaire was employed in this study [18].

Age, gender, residence location, ownership of durable (for calculation of wealth index and quintiles), and parental education were the main independent variables. Age was divided into three categories: (i) 2–2.9 years; (ii) 3–3.9 years; and (iii) 4–4.9 years. There were two types of residential areas: urban and rural. The family wealth distribution was divided into five quintiles: quintile 1 (poorest) to 5 (richest). Parental education was divided into three categories: (i) primary education (primary school or below); (ii) secondary education (secondary school); and (iii) post-secondary education (beyond secondary school). The other variables were children’s books possession (three or more children’s books), toy play (with two or more sorts of toys: homemade toys, shop/manufactured toys, or domestic objects), and electronic device play [19].

The major dependent variables were six parental interactions with their children in the last three days; these were: (a) book reading, (b) storytelling, (c) singing, (d) identifying or counting or drawing, (e) outdoor play, and (f) family play. We have categorised six interactions into two groups; children had parents engaged in at least four out of six interactions as adequate group, and less than four interactions as inadequate group. This is in line with MICS classification and reporting [19,20]. Of these six activities, we further classified into two groups as physical and non-physical interactions. The non-physical interaction included either book reading, storytelling, singing, or identifying or counting or drawing, while the physical play covered either outdoor play or family play. 

### 2.3. Data Analysis 

The investigation was broken down into three stages. First, descriptive statistics were used to give a summary of the data. Second, a Chi-square test was used to examine the relationship between each covariate and parental interactions in the univariable analysis. Finally, to account for the impact of all factors at the same time, multivariable logistic regression was used to determine the association of having parental interactions with the independent variables. Covariates with a *p*-value of less than 0.05 in the univariable analysis were included in the multivariable analysis. The results were presented in the form of adjusted odds ratios (AOR) with a 95% confidence interval (CI). We utilised STATA software version 17 (serial license number: 401709350741) for data analysis.

## 3. Results

### 3.1. Baseline Characteristics and Parental Interactions 

This study enrolled a total of 8856 children under the age of five. As demonstrated in Table 1, most participants, 90%, had at least four interactions with parents. Samples aged 2, 3, and 4 years, had at least four interactions with parents at 90%, 91%, and 90%, respectively. Parents of boys and girls reported similar levels of interactions, with 90% and 91%, respectively. Children in urban areas showed considerably higher parental interactions of at least four interactions than children in rural areas, with 93% and 89%, respectively (*p*-value < 0.001). Children from households with poorer wealth quintiles exhibited significantly lower parental interactions than those from richer families, with 85%, 89%, 91%, 94%, and 96%, respectively, for family wealth quintiles 1 (poorest) to 5 (richest) (*p*-value < 0.001). Furthermore, children living with parents with lower education levels had significantly lower parental interactions than children living with parents with higher education levels (*p*-value < 0.001).

When disaggregating parental interactions by type, family play and outdoor play were the most common parental interactions (about 97–98%) as shown in Figure 1. 

By grouping six types of childhood play into non-physical and physical parental interactions, almost all participants had both types of parental interactions, as shown in Table 2. Children living in urban areas had slightly higher non-physical parental interactions than rural counterparts, despite a small margin of 98% and 97%, respectively (*p*-value < 0.01). Children from families in the poorest wealth quintiles had significantly lower non-physical parental interactions (96%) than those with the richest quintiles (99%) (*p*-value < 0.001). Furthermore, children raised by parents with lower levels of education had considerably lower physical and non-physical parental interactions than children raised by parents with higher levels of education.

### 3.2. Children’s Books Possession, Toy Play, and Electronic Device Play, and Parental Interactions 

Among all children in this survey, approximately 38% (3383/8856) had three or more children’s books, 92% (8087/8814) played toys (with two or more types of toys), and 69% (5641/8842) played electronic devices, as shown in Table 3. Children who had children’s books played with toys, and played with electronic devices, had higher parental interactions at 97%, 91%, and 92%, respectively. We found statistically significant positive associations between these variables and each type of parental interaction.

By grouping six types of parental interactions into physical and non-physical parental interactions, participants possessing children’s books significantly had a 99%- in having both parental interactions, as presented in Table 4. Likewise, children who played with toys significantly had higher physical (99%) and non-physical parental interactions (98%). For the electronic devices play, children who played them had significantly higher physical and non-physical parental interactions, however, a statistical significance was found only in the non-physical parental interactions. 

### 3.3. Screen Time and Parental Interactions 

High screen time was significantly associated with a lower percentage of overall parental interaction, book reading (*p*-value < 0.05), storytelling (*p*-value < 0.01), and singing (*p*-value < 0.05), as shown in Figure 2.

### 3.4. Participants’ Profiles and Parental Interactions: Multivariable Analysis 

We included variables that had a statistical significance from univariable analysis in the multivariable analysis. These are residential area, family wealth index, parental education level, children’s books possession, toy play, and screen time. Based on the multivariate logistic regression analysis, children raised by parents with secondary or post-secondary educations had a significantly greater chance of having parental interactions of at least four activities than children raised by patients who completed primary education (AOR = 1.66, 95% CI: 1.29–2.14 for secondary education, and AOR = 2.34, 95% CI: 1.68–3.24 for post-secondary education)—as displayed in Table 5. Those who had children’s books or played toys significantly had higher parental interactions (AOR = 3.08, 95% CI: 2.37–3.99 for children’s books possessing, and AOR = 1.50, 95% CI: 1.08–2.08 for toy play). Children who spent 1–3 h per day in front of a screen showed a significantly lower chance of having parental interactions than those spending less screen time (AOR = 0.67, 95% CI: 0.55–0.82).

## 4. Discussion

UN Member States committed to SDG target 4.2; this means by 2030, to ensure that all girls and boys have access to quality early childhood development, care, and pre-primary education so that they are ready for primary education [20]. This study monitors progress towards SDG indicator 4.2.3, which measures the percentage of children under 5 years experiencing positive and stimulating home learning environments [20]. 

Our analysis of MICS6 found that in Thailand, approximately 90% of children under the age of five had involved in four or more play activities with parents. The findings were in line with a recent UNICEF report on early child development studies, which ranked Thailand 11th out of 84, mostly low- and middle-income countries across five geographical regions [20]. UNICEF reports 80.8 million children ages three and four years old have low cognitive and/or socio-emotional development in low- and middle-income countries, with the highest prevalence in sub-Saharan Africa, followed by South Asia, and then East Asia and Pacific region [20]. 

Our study also shed light on factors associated with parental interactions among Thai children under the age of five. The children’s books possessing had the greatest degree of association with parental interactions, followed by parental education and the experience of toy play. Longer periods of screen time, in contrast, presented a negative association with parental interactions.

The results corroborate the findings from 2006, 2012, and 2016 MICS, which found that Thai children under the age of five whose parents had higher education levels engaged in more parental interactions (81%, 96%, and 97%, respectively) [15,16,17]. Higher parental education likely creates parental awareness and responsiveness to the children’s natural needs and demand for plays [21,22]. Moreover, educated parents seem to have the ability to obtain specific information about their children’s health requirements [21,22]. Higher education levels of parents also help increase parental involvement in child-rearing practice, and the time used with children is likely to be involved with a more developmentally appropriate academic-related activity [21,22]. Parents with higher education tended to discourage their children from engaging in sedentary activities, such as watching television for too long [23]. Furthermore, parents with lower education levels may undervalue their involvement in the demonstration, full engagement, and family practices [9,11,21,22] in intellectual activities, such as book reading, storytelling, and singing. In addition, emerging parental responsibilities, specifically an increased female labour force participation, reduced time spent playing with children, particularly among low-education and low-income families [24,25,26,27,28].

Apart from 2006, 2012, and 2016 MICS reports, this study further discovered that parental interactions were greatly related to the possession of children’s books, toy play, and electronic device play. This can be explained by the nature of book reading, which usually fosters interaction between parents and children and the likelihood that children adapt some knowledge learned from the book to different forms of play [29,30]. Young children are naturally attracted by animals that frequently appear in children’s novels [29]. Novels with fantastical elements can encourage youngsters to engage in imaginative role-playing [31,32]. Focusing on toys, children can use toys to improve reasoning, physical coordination, and creative thinking skills. Toy exploration promotes the development of cognitive skills such as pretending, cause and effect, problem-solving, and a variety of other executive functions [33,34]. There is a range of different play styles in that children can engage with toys. The older the children, the more creative and sophisticated in playing they become [35]. Besides, some parents tend to provide educational materials for their children [4], this can explain why toys are associated with more engagement in childhood play. Providing inexpensive children’s books and toys (e.g., handmade or ordinary household items) can be considered an excellent option to engage children and parents to spend time together and lead to better childhood development [2,13].

In the case of electronic devices, children demonstrated high levels of engagement during digital play. Electronic books with thoughtful enhancements, such as animated pictures, music, and sound effects support children to engage in content learning and may inspire them to diversify different forms of play [36,37]. However, there is a caveat of electronic devices as children’s mental effort may be focused on interacting with the features rather than paying attention to their parents and surroundings [23]. The finding of this study revealed that shorter screen duration was strongly associated with higher parental interactions. A likely explanation is less screen time provides more opportunities for children to interact with their parents through other types of plays, or external environments, all of which help promote their imaginative skills and developments. Constant stimulation and absorption of visual content on screens also affect young children’s attention span and focus [38]. As a result, the WHO recommends no screen time for children under the age of 2, and no more than one hour a day for those aged 2 to 4 years [39]. Excessive screen time impinges on children’s ability to develop optimally [40]. Moreover, parents should ensure that the content with which their children interact through electronic devices is free of violence and appropriate for their age. Co-viewing and co-playing during screen time are recommended to foster connections and conversations among family members [14,30]. Having dialogues with children while playing, reading, or using materials is advisable. Parents can help promote children’s development by paying attention to content-related discourse, by prompting them to express their thinking, or by frequently asking questions [30,41].

There were several strengths in this study. Firstly, a rigorous national representative sampling technique was conducted. Secondly, a large number of participants were recruited throughout the country. Thirdly, associations between parental interactions and demographic profiles of children were performed through statistical analysis which has not been done in the previous or present report of MICS. Fourthly, a multivariable analysis was applied to control several covariates at the same time. This helps minimise the bias in the estimated results, which might be due to confounders. Despite the study’s merits, several limitations remain. To begin, the sampling technique was based on the Department of Provincial Administration’s household registry. Although this is a common procedure employed by all NSO surveys, it does have a disadvantage as persons without a civil registry number, especially disadvantaged populations, such as homeless people and slum residents, were likely to be excluded from the sampling frame. As a result, the results may overestimate the parental interactions because the sample households exclude the marginalised groups. Secondly, the MICS employs a limited set of parental interactions. For example, out of six activities, four focus on cognitive-based rather than physical-based interactions. Other forms of physical-based interactions i.e., dancing, or indoor exercising, can be added to the survey. Thirdly, additional characteristics such as the quality and type of children’s books and technological devices, as well as the household and community environments, were not captured by the questionnaire. Fourthly, parents or caregivers reporting of their interactions with their children is subject to recall biases and social desirability bias [42]. This bias is a tendency to underreport socially undesirable attitudes and behaviours and to over-report more desirable attributes. Thus, the interpretation of these findings needs careful consideration.

Although this study relies on the MICS dataset which does not allow direct measurement of the frequency and profile of “child play”; parental reporting of interactions with their children through various activities can support child development [43]. We recommend further primary research to assess the prevalence and profile of actual play and child development. Besides, further studies that intend to standardise the amount of play and indicate the degree of the availability of toys or play-related accessories (books, toys, etc.) will be extremely useful and this will help extend the academic value in the field of health promotion in children. 

We continue to monitor progress of SDG target 4.2.3 when the database of the next round of MICS to be conducted by NSO is released to International Health Policy Programme of the Ministry of Public Health.

## 5. Conclusions

This study shows a high proportion of children, 90%, were reported to be engaged with their parents during playtime. The most significant influence on parental interactions was the children’s books possession, followed by parental higher education and the experience of toy play. A long period of screen time presented a detrimental impact on parental interactions. The future of a nation depends on the potential of the young generation developed through, among others, appropriate parental interactions and engagement during their childhood period. We recommend national policymakers, parents, school, and communities actively promote parental interactions through play; and not allow screen time for more than one hour a day for children 2–4 years old. The government and local authorities should provide sufficient supports to children, especially those with lower educated parents and poorer families to ensure an optimal home and community environment, which can significantly enhance children’s development and well-being. We propose that the next MICS survey include more questions on physical-based parental interactions, characteristics of children’s books, and household context. Further primary research to assess the prevalence and profile of actual play and child development is also recommended. 

## Figures and Tables

**Figure 1 ijerph-19-03418-f001:**
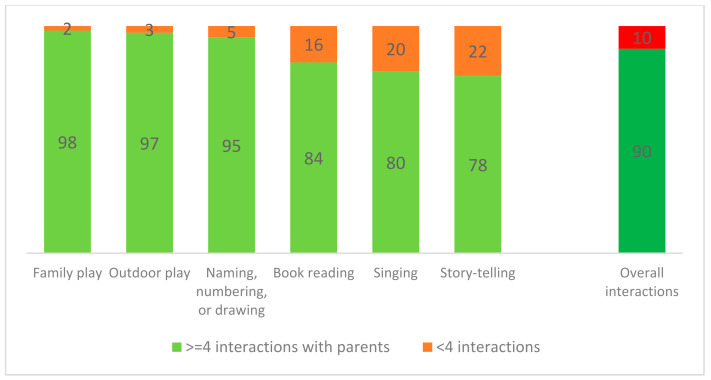
Percentage of the parental interactions by type.

**Figure 2 ijerph-19-03418-f002:**
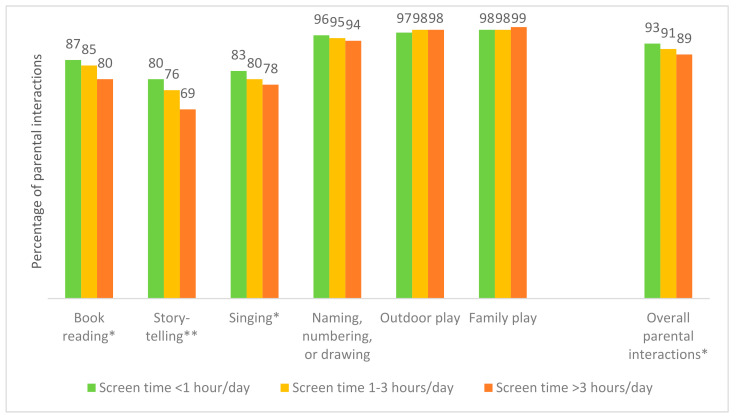
Percentage of the parental interactions by screen time duration. * *p*-value < 0.05, ** *p*-value < 0.01 (for each type of parental interactions and each range of screen time duration).

**Table 1 ijerph-19-03418-t001:** Comparing the parental interactions by children’s attributes.

Variables	At least Four Interactions with Parents (%)	Fewer than Four Interactions with Parents (%)
Total	7995 (90.3)	861 (9.7)
Age group (years)		
2	2754 (89.7)	315 (10.3)
3	2819 (90.7)	290 (9.3)
4	2422 (90.4)	256 (9.6)
Gender		
Male	4070 (89.9)	458 (10.1)
Female	3925 (90.7)	403 (9.3)
Residential area *		
Urban	2870 (92.5)	234 (7.5)
Rural	5125 (89.1)	627 (10.9)
Family wealth (quintiles) *		
1	1879 (85.1)	329 (14.9)
2	1817 (89.0)	225 (11.0)
3	1687 (91.0)	166 (9.0)
4	1508 (93.8)	99 (6.2)
5	1104 (96.3)	42 (3.7)
Parental education level * (*n* = 8518)		
Primary education	4023 (87.6)	572 (12.4)
Secondary education	1821 (93.6)	125 (6.4)
Post-secondary education	1902 (96.2)	75 (3.8)

* *p*-value < 0.001. Note: percentage of parental interactions per all children in each subgroup of a variable.

**Table 2 ijerph-19-03418-t002:** Comparing physical and non-physical parental interactions by children personal attributes.

Variables	Physical Parental Interactions (%)	Non-Physical Parental Interactions (%)
Total	8769 (99.0)	8632 (97.5)
Age group (years)		
2	3039 (99.0)	2988 (97.4)
3	3081 (99.1)	3026 (97.3)
4	2649 (98.9)	2618 (97.8)
Gender		
Male	4488 (99.1)	4413 (97.5)
Female	4281 (98.9)	4219 (97.5)
Residential area		
Urban	3081 (99.3)	3046 (98.1) *
Rural	5688 (98.9)	5586 (97.1) *
Family wealth (quintiles)		
1	2178 (98.6)	2118 (95.9) **
2	2025 (99.2)	1976 (96.8) **
3	1838 (99.2)	1814 (97.9) **
4	1593 (99.1)	1588 (98.8) **
5	1135 (99.0)	1136 (99.1) **
Parental education level (*n* = 8518)		
Primary education	4539 (98.8) *	4441 (96.7) **
Secondary education	1932 (99.3) *	1920 (98.7) **
Post-secondary education	1968 (99.5) *	1960 (99.1) **

* *p*-value < 0.01, ** *p*-value < 0.001 (for each type of parental interaction and each variable). Note: Percentage of each type of parental interaction per all children in each subgroup of a variable.

**Table 3 ijerph-19-03418-t003:** Children’s books possession, toy play, electronic device play, and parental interactions.

Variables	Parental Interactions						
	Book Reading (%)	Storytelling (%)	Singing (%)	Naming, Numbering, or Drawing (%)	Outdoor Play (%)	Family Play (%)	Overall (%)
Children’s books possession							
Yes (*n* = 3383)	2958 (87.4) *	2897 (85.6) *	2897 (85.6) *	3310 (97.8) *	3314 (98.0) *	3330 (98.4) *	3269 (96.6) *
No (*n* = 5473)	3967 (72.5) *	4198 (76.7) *	4198 (76.7) *	5057 (92.4) *	5272 (96.3) *	5345 (97.7) *	4726 (86.4) *
Toy play							
Yes (*n* = 8087)	6868 (84.9) *	6387 (79.0) *	6546 (80.9) *	7674 (94.9) *	7868 (97.3) *	7945 (98.2) *	7971 (91.2) *
No (*n* = 727)	566 (77.9) *	508 (69.9) *	517 (71.1) *	653 (89.8) *	677 (93.1) *	693 (95.3) *	590 (81.2) *
Electronic device play							
Yes (*n* = 5641)	4856 (86.1) *	4418 (78.3) *	4593 (81.4) *	5377 (95.3) *	5484 (97.2) *	5544 (98.3) *	5174 (91.7) *
No (*n* = 3201)	2599 (81.2) *	2495 (77.9) *	2491 (77.8) *	2977 (93.0) *	3088 (96.5) *	3117 (97.4) *	2809 (87.8) *

* *p*-value < 0.001 (for each parental interaction and each variable). Note: percentage per all children in each subgroup of a variable.

**Table 4 ijerph-19-03418-t004:** Children’s books possession, toy play, and electronic device play, and having two types of parental interactions.

Variables	Having Physical Parental Interactions (%)	Having Non-Physical Parental Interactions (%)
Children’s books possession		
Yes (*n* = 3383)	3361 (99.4) *	3353 (99.1) **
No (*n* = 5473)	5408 (98.8) *	5279 (96.5) **
Toy play		
Yes (*n* = 8087)	8024 (99.2) **	7910 (97.8) **
No (*n* = 727)	704 (96.8) **	680 (93.5) **
Electronic device play		
Yes (*n* = 5641)	5593 (99.2)	5524 (97.9) **
No (*n* = 3201)	3162 (98.8)	3095 (96.7) **

* *p*-value < 0.01, ** *p*-value < 0.001 (for each type of parental interactions and each variable). Note: percentage per all children in each subgroup of a variable.

**Table 5 ijerph-19-03418-t005:** Multivariable analysis of having parental interactions of at least four activities.

Variables	Adjusted Odds Ratio	95% Confidence Interval
Residential area		
Rural	0.91	0.73–1.13
(ref = urban)		
Family wealth		
Quintile 2	0.99	0.75–1.32
Quintile 3	0.96	0.72–1.29
Quintile 4	1.15	0.83–1.60
Quintile 5	1.20	0.77–1.87
(ref = quintile 1)		
Parental education level		
Secondary education	1.66 **	1.29–2.14
Post-secondary education	2.34 **	1.68–3.24
(ref = primary education)		
Children’s books possession		
Yes	3.08 **	2.37–3.99
(ref = no)		
Toy play		
Yes	1.50 *	1.08–2.08
(ref = no)		
Duration of screen time		
1–3 h/day	0.67 **	0.55–0.82
>3 h/day	0.61	0.33–1.11
(ref = <1 h/day)		

* *p*-value < 0.05, ** *p*-value < 0.001.

## Data Availability

The data presented in this study are available at https://mics.unicef.org/surveys (accessed on 14 February 2022).

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
