# Peer review of "Self-Reported Parental Interactions through Play with Young Children in Thailand: An Analysis of the 2019 Multiple Indicator Cluster Survey (MICS)"

_ijerph, 2022, doi:10.3390/ijerph19063418_

Round 1
Reviewer 1 Report
The study is potentially interesting. Unfortunately the conceptualisation and measurement of play is problematic. The instrument used is not a validated measure of play and as far as I can tell, it wasn’t intended to be used to measure play.
The Introduction provides a very basic understanding of play as a construct and no clear theoretical position on play. The authors opt to conceptualise play as described by United Nations Children’s Fund. This conceptualisation has no relationship to the measurement of play. The measurement of play is based on children engaging in a) book reading, b) story-telling, c) singing, d) identifying or counting or drawing, e) outdoor play, and f) family play, with parental report of the presence of 4 of these indicating ‘adequate play’.
I checked the MICS for Thailand and found the following in MICS6.UF.6
*************************************************
In the past 3 days, did you or any household
member age 15 or over engage in any of the following
activities with (name):
If ‘Yes’, ask:
Who engaged in this activity with (name)?
A foster/step mother or father living in the household
who engaged with the child should be coded as
mother or father.
Record all that apply.
‘No one’ cannot be recorded if any household member
age 15 and above engaged in activity with child.
MOTHER FATHER OTHER NO ONE
[A] Read books or looked at picture
books with (name)?
[B] Told stories to (name)?
[C] Sang songs to or with (name),
including lullabies?
[D] Took (name) outside the home?
[E] Played with (name)?
[F] Named, counted, or drew things
for or with (name)?
*********************************************************
The MICS items used relate to stimulation of the child by adults in various activities, but are not specifically about play. I was also unsure how use of 4/6 indicated that this was adequate as these don’t seem to be quantified other than occurring in the past 3 days. Furthermore, it is possible that a parent would report 4/6 of these options, but the child is never engaged in play. It is also possible that a parent would report less than 4 of these 6 items, but the child has a lot of opportunities to play e.g. spends a lot of time with peers playing imaginative games.
I haven’t reviewed further as I don’t see how the manuscript could be revised given the difficulties with measuring play. I would note though that the Introduction and Discussion lack the rigor generally expected for reporting in a high ranking refereed journal.
I appreciate the difficulties of using data collected for a different purpose. Perhaps there are other ways to analyse the data that align with the constructs measured in the MICS.
Author Response
The study is potentially interesting. Unfortunately the conceptualisation and measurement of play is problematic. The instrument used is not a validated measure of play and as far as I can tell, it wasn’t intended to be used to measure play.
Response
Thank you for your comment. We have revised the objective of the study to assess the prevalence of parental interactions with their children and identify household characteristics that influence the level of parental interactions with their children under the age of five (page 3, lines 102-105).
The Introduction provides a very basic understanding of play as a construct and no clear theoretical position on play. The authors opt to conceptualise play as described by United Nations Children’s Fund. This conceptualisation has no relationship to the measurement of play. The measurement of play is based on children engaging in a) book reading, b) story-telling, c) singing, d) identifying or counting or drawing, e) outdoor play, and f) family play, with parental report of the presence of 4 of these indicating ‘adequate play’.
Response
Thank you and we agree with your comment. Our revised manuscript does not address “Play” as such. Instead, we analyze the self-reported parental interactions with their children in six dimensions (a) book reading, b) story-telling, c) singing, d) identifying or counting or drawing, e) outdoor play, and f) family play) based on the questions applied by the MICS. We thought that although we do not address “Play” as such, the parental interactions with their under five years old children are critical, and not many studies address this.
We have conducted a further review of the literature and a paragraph was introduced in the introduction section as followed:
A systematic review suggests that due to the complexity of dyadic interactions, there are both positive and negative outcomes on child development [5]. Stimulating play also enhances the adaptability of a child to a chronic somatic condition such as cystic fibrosis, congenital heart defects, and promotes cognitive, social, emotional, and psychomotor functioning; who otherwise will have a significantly increased risk for physical, social, emotional, and cognitive problems later in life [6]. Though loose parts plays are promising, their impacts on children's cognitive, social and emotional development need further research [7].
I checked the MICS for Thailand and found the following in MICS6.UF.6
************************************************
In the past 3 days, did you or any household
member age 15 or over engage in any of the following
activities with (name):
If ‘Yes’, ask:
Who engaged in this activity with (name)?
A foster/step mother or father living in the household
who engaged with the child should be coded as
mother or father.
Record all that apply.
‘No one’ cannot be recorded if any household member
age 15 and above engaged in activity with child.
MOTHER FATHER OTHER NO ONE
[A] Read books or looked at picture
books with (name)?
[B] Told stories to (name)?
[C] Sang songs to or with (name),
including lullabies?
[D] Took (name) outside the home?
[E] Played with (name)?
[F] Named, counted, or drew things
for or with (name)?
*********************************************************
The MICS items used relate to stimulation of the child by adults in various activities, but are not specifically about play. I was also unsure how use of 4/6 indicated that this was adequate as these don’t seem to be quantified other than occurring in the past 3 days. Furthermore, it is possible that a parent would report 4/6 of these options, but the child is never engaged in play. It is also possible that a parent would report less than 4 of these 6 items, but the child has a lot of opportunities to play e.g. spends a lot of time with peers playing imaginative games.
Response
Thank you and we do agree with your thoughtful comments.
This study is an analysis of the frequency and profiles of parental interactions using MICS. Hence, the findings are guided by the questions asked. We agree there might be mismatches between a high level of parental report of interactions, and a low level of actual play among children, and vice versa.
To address this, in the last paragraph of the discussion, we introduced a text on the limitation of parental reporting of their interaction with children; that the frequency may not associate with actual play among children, and vice versa, as follows (page 12, lines 361-366).
“Fourthly, parents or caregivers reporting of their interactions with their children is subject to recall biases and social desirability bias [41]. This bias is a tendency to underreport socially undesirable attitudes and behaviors and to over-report more desirable attributes. Thus the interpretation of these findings needs careful consideration.”
I haven’t reviewed further as I don’t see how the manuscript could be revised given the difficulties with measuring play. I would note though that the Introduction and Discussion lack the rigor generally expected for reporting in a high ranking refereed journal.
I appreciate the difficulties of using data collected for a different purpose. Perhaps there are other ways to analyse the data that align with the constructs measured in the MICS.
Response
Thank you for your comment. We have a major revision of our manuscript, not addressing “play” as such as the tools used by MICS are not intended to measure play, but re-orient our paper as parental interactions with their children using various activities. We introduced a new paragraph at the end of the discussion as follows.
“Although this study relies on MICS dataset which does not allow direct measure-ment of the frequency and profile of “child play”; parental reporting of interactions with their children through various activities can support child development [42]. We recommend further primary research to assess the prevalence and profile of actual play and child development. Besides, further studies that intend to standardise amount of play and indicate the degree of the availability of toys or play-related accessories (books, toys, etc.) will be extremely useful and this will help extend the academic value in the field of health promotion in children.”
For English editing, thank you for your comment. We have corrected grammatical errors throughout the manuscript.
Reviewer 2 Report
Congratulations to the Authors of the interesting research results. Taking into account the fact that the research was conducted in Thailand, among respondents from various social groups, the authors skilfully showed significant differences in the play of young children. The results of this research may also be interesting for readers from other parts of the world. I believe that researchers in developing countries are getting similar results. The authors made interesting conclusions and pointed to the negative impact on the play of children who use electronic devices excessively (smartphones, tablets, computers and TV).
- The authors presented the results of research on the role of play among children under 5 in Thailand.
- In terms of methodology, the text was properly prepared. I highly appreciate the applied research procedure.
- The aim of the conducted research was to identify factors influencing the development of cognitive functions in children. The authors used in-depth interviews in the research. It is worth emphasizing that the interviews were conducted with mothers or legal guardians of children from various social groups and with different socio-economic status.
The authors in the first part stated that there is no universal definition of play. I think it is worth recalling the results of research by the famous developmental psychologist David Whitebeard (1948-2021). I suggest for example: Whitebread, D. (2015). Childhood in crisis: the loss of play. Cambridge Primary Review Trust blog, available at: http://cprtrust.org.uk/cprt-blog/crisis-in-childhood/#comment-7049
The text requires some editorial corrections:
page 1 - "21st-century skills - and quotation marks;
page 2 - box 1 - children - lowercase; at the end of each sentence - dot;
I recomend the manuscript for publication.
Author Response
Congratulations to the Authors of the interesting research results. Taking into account the fact that the research was conducted in Thailand, among respondents from various social groups, the authors skilfully showed significant differences in the play of young children. The results of this research may also be interesting for readers from other parts of the world. I believe that researchers in developing countries are getting similar results. The authors made interesting conclusions and pointed to the negative impact on the play of children who use electronic devices excessively (smartphones, tablets, computers and TV).
- The authors presented the results of research on the role of play among children under 5 in Thailand.
- In terms of methodology, the text was properly prepared. I highly appreciate the applied research procedure.
- The aim of the conducted research was to identify factors influencing the development of cognitive functions in children. The authors used in-depth interviews in the research. It is worth emphasizing that the interviews were conducted with mothers or legal guardians of children from various social groups and with different socio-economic status.
- The research results have been duly analyzed.
Response
Thank you for your comment.
The authors in the first part stated that there is no universal definition of play. I think it is worth recalling the results of research by the famous developmental psychologist David Whitebeard (1948-2021). I suggest for example: Whitebread, D. (2015). Childhood in crisis: the loss of play. Cambridge Primary Review Trust blog, available at: http://cprtrust.org.uk/cprt-blog/crisis-in-childhood/#comment-7049
Response
Thank you for your suggestions. We have reviewed the reference, and the related literature you have suggested. We have introduced the introduction section (page 2, lines 65-69):
“In the UK and elsewhere, modern, urbanization and technologies have changed child play experiences of free play and unstructured time. Around half of all children worldwide now live in urban settings, and experience very reduced opportunities for outdoor free play in natural environments, due to parental concerns about traffic, ‘stranger danger’, germs, and disease [8].”
The text requires some editorial corrections:
page 1 - "21st-century skills - and quotation marks;
Response
Thank you for your comment. We have revised accordingly.
page 2 - box 1 - children - lowercase; at the end of each sentence - dot;
Response
Thank you for your comment. We have revised accordingly.
I recomend the manuscript for publication.
Response
Thank you for your comment.
Response
For English editing, thank you for your comment. We have corrected grammatical errors throughout the manuscript.
Reviewer 3 Report
This is an interesting study using a nationally representative sampling technique and a large number of children. I have some comments especially for the methods and results, hoping to strengthen this study.
I expect more information for section: 2.2 Data Collection, Questionnaire Design, and Variable Management.
Line 115-117, the authors may provide more information on how were the survey questions designed. Why 3 or more books indicated availability? How did you ask about the availability of electronic devices? The term electronic device is very broad, does it include TV and any kinds of computer or mobile technology? Also, it is surprising that book and electronic devices were unavailable over 5000 and 3000 families, respectively.
Line 119, how was the adequate level of play (4 out of 6 types of play activities) determined? Do you have any references?
In the results, I’m confused about the data presentation, please clarify the followings:
In Table 1, it seems that the ‘Adequate level of play (%)’ and ‘Inadequate level of play (%)’ are presenting % out of each category of overall children. For overall children, all % are 100 which looks redundant. Suggest removing % for overall children, and indicate in the footnote that the % for Adequate/ Inadequate level of play are the percentages under each category of overall children.
Line 174, what analyses you have performed to find statistically significant positive associations between availability of this equipment and higher adequate levels of play in each type of six play activities?
In Tables 2, 3, and 4, it’s not clear that the *marker (p value < 0.001) indicated what results from what analyses.
By looking at the values (%) of physical and non-physical played children in Table 4, the available and unavailable children presented pretty similar patterns. However, results in the table 4 indicated significant differences (*p value < 0.01, **p value < 0.001) in every cluster except for physical played children in comparing availability of electronic devices. The authors need to provide more descriptions about the results in the text.
In discussion, the authors indicated that the results corroborate the findings from the 2006, 2012, and 2016 MICS. I suggest highlighting new findings in the following paragraphs to tell the readers what’s new in the 2019 MICS.
Author Response
This is an interesting study using a nationally representative sampling technique and a large number of children. I have some comments especially for the methods and results, hoping to strengthen this study.
I expect more information for section: 2.2 Data Collection, Questionnaire Design, and Variable Management.
Line 115-117, the authors may provide more information on how were the survey questions designed. Why 3 or more books indicated availability? How did you ask about the availability of electronic devices? The term electronic device is very broad, does it include TV and any kinds of computer or mobile technology? Also, it is surprising that book and electronic devices were unavailable over 5000 and 3000 families, respectively.
Response
Thank you for your comment. The questionnaire was constructed according to the MICS global standardized format which facilitates international comparisons. We have provided more accurate details of the questions as follows (pages 3-4, lines 137-141).
“The other variables were children’s books possession (3 or more children's books), toy play (with two or more sorts of toys: homemade toys, shop/manufactured toys, or domestic objects), and electronic device play.”
We also agree that the classification of availability of books or any device does not have a standard reference. This also points to further study in this field, see page 12, lines 373-376.
Line 119, how was the adequate level of play (4 out of 6 types of play activities) determined? Do you have any references?
Response
Thank you for your comment. In line with Reviewer 1 comments and suggestions; we have reformulated the focus of the paper; instead of “play’ we use “self-reported parental interactions with their children. This is because the questions in the MICS ask about the frequency and profile of parental interaction; without direct measurement of actual play in children.
We have categorised six interactions into two groups as children had parents engaged in at least four out of six interactions as adequate, and less than four interactions as inadequate. This is in line with MICS classification and reporting (page 4, lines 148-151).
In the results, I’m confused about the data presentation, please clarify the followings:
In Table 1, it seems that the ‘Adequate level of play (%)’ and ‘Inadequate level of play (%)’ are presenting % out of each category of overall children. For overall children, all % are 100 which looks redundant. Suggest removing % for overall children, and indicate in the footnote that the % for Adequate/ Inadequate level of play are the percentages under each category of overall children.
Response
Thank you for your comment. We have revised accordingly (page 4, line 184). We have also provided footnotes for tables 2, 3, and 4.
Line 174, what analyses you have performed to find statistically significant positive associations between availability of this equipment and higher adequate levels of play in each type of six play activities?
Response
Thank you for your comment. a Chi-square test was used to examine the relationship between each covariate and parental interactions in the univariable analysis as indicated in the section 2.3 data analysis (page 4, lines 158-160).
In Tables 2, 3, and 4, it’s not clear that the *marker (p value < 0.001) indicated what results from what analyses.
Response
Thank you for your comment. The p-value was determined for each parental interaction and each variable. We have provided an explanation in the footnote of each table.
By looking at the values (%) of physical and non-physical played children in Table 4, the available and unavailable children presented pretty similar patterns. However, results in the table 4 indicated significant differences (*p value < 0.01, **p value < 0.001) in every cluster except for physical played children in comparing availability of electronic devices. The authors need to provide more descriptions about the results in the text.
Response
Thank you for your comment. We have corrected the p-value and added texts on page 7, lines 232-235.
“For the electronic devices play, children who played them had significantly higher physical and non-physical parental interactions, however, a statistical significance was found only in the non-physical parental interactions.”
In discussion, the authors indicated that the results corroborate the findings from the 2006, 2012, and 2016 MICS. I suggest highlighting new findings in the following paragraphs to tell the readers what’s new in the 2019 MICS.
Response
Thank you for your comment. We have added texts to highlight new findings (paragraphs 3 and 4 of the discussion part) at the beginning of the third paragraph (from line 296, page 10). We have also added the strengths of the study in the discussion to reiterate the new findings (page 11, lines 336-343).
Response
For English editing, thank you for your comment. We have corrected grammatical errors throughout the manuscript.
Round 2
Reviewer 3 Report
The authors' responses and revisions are fine to move on to publication.
Author Response
Dear Reviewer,
Thank you for your valuable comments. We’ve revised the manuscript as suggested as follows.
- English language and style are fine/minor spell check required.
Thank you for the comment. The whole manuscript was checked for misspellings and grammatical errors.
